# Gendered perspectives of yoga in the Key Stage 1 classroom: Qualitative content analysis indicates contrasting views of teachers and pupils

Katie Wilkin *, Claire Thornton, Georgia Allen-Baker

Faculty of Health and Wellbeing, Northumbria University, Newcastle Upon Tyne, United Kingdom

* katie.wilkin@northumbria.ac.uk

## Abstract

The gendering of physical activities is ubiquitous, with those involving strength, endurance, and physical contact considered masculine, and those involving concentration, presentation, and flexibility considered feminine. Yoga, for instance, tends to be regarded by adults and adolescents as a feminine activity for women/ girls, however, it is not known whether younger children share this view. Using data from six individual teacher interviews and a child-friendly task with 23 pupils (working in four separate groups), qualitative content analysis (QCA) was used to identify and explore the gendered perspectives of yoga held by Key Stage 1 teachers and pupils in schools across the North East of England. Data were considered according to Gender Schema Theory (GST) and indicate that, although young pupils seem to hold no consensus view of yoga being a female/ feminine activity, teachers observe reluctance from some of their male pupils during yoga activities in school. As previous research has revealed that teachers generally expect male pupils to be more competitive than their female peers, the findings are discussed in relation to the non-competitive nature of yoga.

## Introduction

Sporting and physical activities are consistently gendered [1], with aesthetic activities requiring concentration, presentation, and flexibility (such as dance and gymnastics) more often considered feminine [2], and activities involving strength, endurance, and physical contact more typically conceived of as masculine [3]. Such gender stereotypical views of physical activity are held by adults [4] – including teachers [5,6], adolescents [7], and primary schoolchildren [8].

As one of the most popular fitness and wellbeing practices worldwide [9], yoga has been described as a global phenomenon [10] but its appeal is not universal. For instance, yoga is heavily marketed towards women [11], and adult participation

**Data availability statement:** The study's data are not being made publicly available without restriction because some participants were very young (5 years of age) and the topic is potentially sensitive (gendering / gender stereotypes). Anonymised transcriptions of the participants' data are available upon request by interested parties with legitimate purpose. Please email Northumbria University's research data management team (as.researchdata@ northumbria.ac.uk) with any queries or access requests.

**Funding:** The author(s) received no specific funding for this work.

**Competing interests:** The authors have declared that no competing interests exist.

rates show that women are far more likely to practice yoga than men [12,13]. Indeed, qualitative studies find that men often deem yoga to be a 'feminine' pursuit [14] and female-dominated activity [15] which discourages male participation [16] and can create feelings of embarrassment and self-consciousness in male participants [17].

Moreover, in yoga intervention studies, men are less likely to volunteer [18] and more likely to withdraw their participation than women [19], and in education settings, male students can be resistant to school-based yoga [20,21] and may feel peer pressure to distance themselves from yoga-based activities [22]. Indeed, evidence suggests that male students are less likely to volunteer for research involving yoga [23] and rate school-based yoga interventions less positively than their female peers [24].

Qualitative studies also suggest that teenage males consider yoga to be "too calm" as it lacks competition [25], and regard yoga as "feminine", "gay" [22] and "for moms", and as such consider yoga inappropriate for boys or men [26]. Indeed, males between ten and 18 years of age can believe they lack the physical capability and flexibility to practice yoga [27]. However, there has been limited research with younger schoolchildren exploring their views on this issue. In fact, we could find only one paper with reference to this [28] which found that some upper grade boys in elementary school (i.e., up to ten years of age) prefer more rigorous activities and sports, and consider yoga to be "uncool" and "feminine" [28].

Yet, yoga is increasingly being offered to children in schools [29], and there is recent evidence of Key Stage 1 teachers (working with pupils between five and seven years of age) delivering such activities in the classroom [30] so it is timely and important to consider younger schoolchildren's views of this activity. Moreover, although studies have reported teachers' perceptions of how pupils generally respond to school-based yoga interventions [31,32] we could find no qualitative study describing teachers' views of any differences in young male and female pupils' engagement with yoga activities. Therefore, this study aims to provide some initial understanding towards two key issues: from teachers' perspectives, how do male and female Key Stage 1 (KS1) pupils engage with school-based yoga, and what are the attitudes and opinions of male and female KS1 pupils towards these activities?

This work was written in accordance with the Standards for Reporting Qualitative Research (SRQR) [33]. The overall methodological approach underpinning this research is pragmatism, and qualitative content analysis (QCA) is the specific methodology used to explore and analyse the data gathered. Literature relating to Gender Schema Theory (GST) was used to guide understanding and interpretation.

N.B., The World Health Organization defines gender as the characteristics of women, men, girls, and boys that are socially constructed, and the term sex as the different biological and physiological characteristics of females, males, and intersex persons [34]. In this paper, we use the term 'gender' to cover both terms as per Resaland et al., 2018 [35].

## Theoretical framework: Gender Schema Theory

Gender Schema Theory (GST, first described by Bem, 1981 [36]) posits that children's internal motivation to conform to societal definitions and expectations of

femininity and masculinity induces them to behave in gender stereotypical ways [37]. According to GST, children develop ideas (gender schemas) about what it means to be masculine or feminine from an early age [38], and these gender schemas are then used to categorise information, make decisions, and regulate behaviour [39].

It is argued that a child develops their sense of self and gender identity largely through behaviours learned and reinforced in schools [40], and there is much evidence that schoolteachers influence children's gender schemas and behaviours [41–44]. Indeed, despite a widespread shift in cultural practices towards an expectation of gender equality, many teachers in the UK continue to endorse traditional gender stereotypes [45,46], including for children's physical abilities [47] and preferences for physical activities [48].

However, it is important to note that some children identify themselves counter to society's norms of presumed sex and gender congruence [49]. Indeed, GST has been criticised as restrictive and marginalising of transgender and/ or nonbinary experiences [50], and novel theories of gender development have been proposed [51], such as Gender Self-Categorisation [52,53], with the aim of advancing gender research to be inclusive and affirming of all individuals and experiences [50].

## Methods

**Positionality.** The lead author identifies as female, is white-British, middle-class, and at the time of writing was middle-aged with one daughter attending first school. Born in Newcastle-upon-Tyne, the lead author speaks with a 'Geordie' North East accent which may have fostered rapport with the participants in this study. She previously trained and worked as a Clinical Associate in Applied Psychology for children and young people, and felt comfortable engaging with the pupil participants to encourage their ideas, opinions, and perspectives. The lead author's interest in this study was shaped by personal positive experiences with yoga and mindfulness, which she practices regularly and finds valuable in her daily life.

The study's co-authors, Georgia Allen-Baker and Claire Thornton, acted as 'critical friends' during data analysis, i.e., they were trusted allies and critics [54], who appraised, elaborated, and enhanced the ideas presented here.

Georgia Allen-Baker is a middle-class, white female, who has two male children of primary school age who have taken part in yoga activities at school. Claire Thornton is a middle-class, white female, who has coached a variety of sports and worked with athletes of diverse abilities from a range of backgrounds.

**Context.** The study is located in the North East of England, one of the most disadvantaged and deprived areas in the country [55]. For instance, school-age children in the North East are more likely to be eligible for Free School Meals (FSM) than their peers in the rest the country [56]. Indeed, in 2025 the average percentage of pupils eligible for FSM in England's schools was 25.7% [57] yet the teachers in this study work in schools with an average of 31.25% pupils eligible for FSM.

**Participants.** Qualitative research typically involves purposeful sampling based on participants' knowledge or experience of phenomena [58]. To recruit participants for the current study, KS1 teachers in state/ non-private schools across England's North East, who previously had completed a survey into their use of classroom-based yoga/ mindfulness [30] and who had consented to be contacted for follow-up, were emailed and asked to consider taking part. Recruitment of teachers took place between 28th July 2024 and 25th September 2024, while pupil recruitment took place between 18th November 2024 and 20th February 2025.

In total, six KS1 teachers, and 23 KS1 pupils participated. All teachers identified as female and worked in different school across the North East (please see Table 1 below). The pupil participants (12 female and 11 male) were five or six years of age and attended two different schools in the region (please see Table 2 below). All participants spoke English as their first language. One pupil's ethnicity was categorised as 'mixed/ multiple – White and Asian', the ethnicity of the other 22 pupils was categorised as 'White British'.

## Procedures

University research ethics approval was granted for qualitative data collection with KS1 teachers (REF: 6792) and KS1 pupils (REF: 7052). For the teacher interviews, participant information sheets were provided directly to teachers, and

**Table 1. Teacher participants.**

| Ppt ID | Ppt ethnic background | Teaching experience | School ID and population* |
|---|---|---|---|
| Teacher 1 | Asian British – Pakistani | 12–14 yrs | S1, Primary. |
| Teacher 2 | White British | 15 + yrs | S2, Infant. |
| Teacher 3 | White British | 15 + yrs | S3, First. |
| Teacher 4 | White British | 12–14 yrs | S4, Primary. |
| Teacher 5 | White British | 6–8 yrs | S5, Primary. |
| Teacher 6 | White British | 3–5 yrs | S6, Primary. |

**Table 2. Pupil participant groups.**

| School ID and population* | Most recent Ofsted rating** | Group ID | N (male or female) |
|---|---|---|---|
| S2, Infant. | Good (2022) | Group 1 | 6 (male) |
| | | Group 2 | 7 (female) |
| S7, First. | Good and Outstanding (2024) | Group 3 | 5 (male) |
| | | Group 4 | 5 (female) |

* Schools providing Key Stage 1 education in the North East of England are either Infant schools (with pupils up to seven years of age), First schools (with pupils up to nine years of age), or Primary schools (with pupils up to 11 years of age). All pupil participants in this study were in 'Year 1' and all were five or six years of age.

** Ofsted is the UK government's Office for Standards in Education, Children's Services and Skills. Ofsted routinely inspects and evaluates services providing education across England

written consent obtained prior to interview. For the pupil qualitative task, Head Teachers provided written gatekeeper consent prior to the distribution of participant information sheets to pupils' parents/ guardians. Parents/ Guardians were advised that they could withdraw their child's involvement without reason at any time, and at the beginning of the task, each participating child's assent was obtained verbally and on paper with child-friendly materials.

All 23 participating pupils engaged and contributed well: no additional prompts or support were needed to encourage pupils to partake in the task. All pupils created a piece of artwork, and offered responses, verbally and/ or physically, at will throughout. However, the pupils' verbalisations were often brief, one- or two-word descriptions of their experiences and judgements, so we obtained less detailed data from these participants than from the teachers, who, as would be expected, provided more detailed and comprehensive responses during interview.

## Data collection

**Pupil qualitative task.** Twenty-three pupils completed the task in four groups: one group of six, one group of seven, and two groups of five. Group sizes complied with adult-to-child ratio guidance from the UK government's Department for Education [59] and the NSPCC [60]. The lead author met the pupils in their classroom with their teacher initially, then each group completed the task with the lead author for approximately 15 minutes in a separate room in their school (a time-limit was necessary to ensure minimal disruption to the school day). Each group session was video recorded to capture audio and visual evidence of the responses of each child, thus ensuring that all contributions could be correctly assigned to each pupil participant.

To begin, each pupil was offered an A4 sheet of white paper with a basic outline of a child – either a 'boy', a 'girl', or a neutral figure. Each child chose the outline they preferred: ten of the 11 male pupil participants chose the 'boy' outline, and 11 of the 12 female pupil participants chose the 'girl' outline for the art task. The remaining two pupils chose to work with

the 'neutral' figure outline. An abundance of coloured crayons and stickers were freely available to use during the task, and the children could sit, stand, and move around the room as they wished, with tables available if/ when a surface was required. Prior to starting, the children were told that they do not *need* to draw or create or indeed talk during the task if they wished not to do so. This was to allow each child to control their level of participation [61] in an effort towards equalising the power imbalance between adult researcher and child participant [62].

To introduce the task, it was explained that the researcher hopes to find out what the children think and feel about the yoga activities they do with their teacher in the classroom. The researcher asked questions to confirm that the children were familiar with and understood the word 'yoga' and emphasised that the children are free to ask questions at any time if anything is unclear. The researcher then invited the children to draw, stick, or write anywhere on their paper to show their thoughts and feelings about doing yoga in class. As the children engaged in the activity, the researcher began positive conversations to encourage verbal description (e.g., "This looks interesting, please tell me about what you have drawn/ written here"; or, "What can you tell me about the yoga you do in class with your teacher?"). Open-ended questions were used wherever possible. Also, 'Why…?' questions were avoided as they imply causation [63] and require metacognitive and metalinguistic skills that young children may not yet possess [64].

As unstructured drawing activities can feel unsafe for some children [65], the methodology involved a structured art task, with predetermined materials, so that each group worked with the same initial stimuli and materials, task introduction, and request from the researcher to provide consistency [66] and to make some effort to control context and motivation across groups.

Upon completion, each child was awarded a certificate and escorted back to their classroom. Subsequently, the pupils' verbal contributions were analysed using the QCA approach (described in more below and in the supplementary material). As there is little guidance regarding how to analyse drawing for meaning, particularly in the field of education [67], with few methods of interpreting visual art, and even fewer explanations of how to apply those methods [68], this study includes no efforts to interpret the children's artwork.

**Teacher interviews.** Interviews aimed to explore teachers' experiences of using yoga with their KS1 pupils, and specifically, whether teachers identify pupils who do, or do not, engage particularly well with these activities. A semi-structured interview schedule was developed to ensure key topics were covered, but interviews were responsive and pursued any relevant issues as they arose [69]. Indeed, at the outset, the gendering of yoga was not highlighted for discussion, but as teachers referred to pupils' gender in relation to engagement with yoga, further information was sought to expand upon this. For instance, one planned interview question was: "Are there any children in your class who do not engage particularly well with the yoga activities?" When one teacher replied by saying, "I mean, some of the boys weren't brilliant at first with joining in" the researcher responded with an unplanned but related follow-up question: "What are your thoughts on that – why do you think some boys weren't initially great at joining in?"

All interviews were recorded and took place via telephone or video-call. Each interview lasted between 35 and 50 minutes and teacher participants chose the date, time, and method (telephone or video-call) according to personal preference and convenience.

## Qualitative Content Analysis (QCA)

The eight steps of QCA (as described by Schreier, 2014 [70]) were followed in order to summarise and reduce the data in relation to our specific research questions, and to the relevant GST literature. The authors met regularly to discuss the process and debate any emerging subcategories and codes, with the aim of challenging assumptions, minimising bias and subjectivity, and improving the reliability of findings. The research team agreed that data saturation had been reached once it was possible to use the data to address the research questions described in the Introduction section, and it was no longer possible to generate further insights from the available recordings and transcripts. Each member of the research team offered varied perspectives during interpretation, with open acknowledgement of each individual's prior experiences

and opinions relating to this research. Consensus was achieved through reflective discussion and critical engagement with the data.

Full details of each step, with examples from the current study, can be found in the supplementary material provided alongside frequency counts to indicate the prevalence of each subcategory and code. A graphic summary of the final QCA coding frame is provided below, as well as a table detailing the QCA main category, subcategories, and codes, with examples from the data. The full coding frame is also provided in the supplementary material. N.B. Teachers' and pupils' data were analysed together to explore and illustrate how the findings from both populations are related.

## Findings and discussion

One main category was used for all selected pupil and teacher data. From this, two subcategories were generated: one with three codes, and one with four codes (please see Table 3 and Fig 1 below).

### Children's views on yoga and gender

**Pupil interest, enjoyment, aptitude, and knowledge of yoga.** During the task, roughly even numbers of boys (n = 9) and girls (n = 10) physically demonstrated yoga poses, with five girls and six boys also providing names for certain poses. In addition, roughly even numbers of girls (n = 3) and boys (n = 4) described also doing yoga activities at home, for instance,

*"In the house… I do it myself"* (Female P7)

*"Sometimes I do it at home"* (Female P10)

*"I do it both. School and my house"* (Male P11)

*"…at home in my special compartment… I do it in the den under the stairs"* (Male P9)

The extension of yoga practice beyond school/ preschool into children's homes is reported elsewhere from studies with six- and seven-year-olds [31,71] as well as three- to five-year-old children [72]. However, as far as we are aware, this is the first study to indicate that female and male KS1 pupils are equally likely to practice yoga at home after taking part in school.

**Gendered perceptions and pupil experiences of yoga.** When prompted to think whether yoga is 'for girls/ for boys/ for everyone', ten pupils (five girls and five boys) responded that yoga is 'for' their gender, which is consistent with research that shows children of this age often show gender ingroup bias [73], for instance often describing their own gender in more positive and favourable terms [74]. Boys in this study made comments such as:

*"It's, it's boys"* (Male P3)

*"Only for boys"* (Male P1)

*"Yeah we're better. Boys… usually have better balance and manage to stay in poses longer than girls. Because girls aren't usually as strong as boys, so [nods], yeah"* (Male P9)

Hence, one male pupil stated that boys are "better" at yoga owing to superior physical skill. However, four female pupils also provided reasons for believing that girls are more skilled at yoga, including:

*"Girls… I don't think boys can do like a cobra, erm, cobra, like put their legs to their head"* (Female P10)

*"Erm, I don't think boys are very good because I don't think boys do, er, lots of clubs"* (Female P9)

**Table 3. QCA table displaying the main category, two subcategories, and seven codes (with frequency counts), and examples from the data.**

| Main category | Subcategory | Code and frequency count | Example/ Quote |
|---|---|---|---|
| Information relating to teachers' and pupils' gendered views of yoga | Pupils' apparent confidence and comfort when demonstrating an interest in, aptitude for, or knowledge of yoga | Pupil physically demonstrates a yoga posture, pose or movement<br>19 pupils<br>10 girls<br>9 boys | Male P8 [prayer hands, one leg balance with other foot on inside knee – tree pose]<br>Female P10 [lying face down on the floor, lifting the shoulders with arched back, legs bent with toes touching head – cobra pose] |
| | | Pupil verbally names a yoga pose, posture or movement<br>11 pupils<br>5 girls<br>6 boys | Female P5 *"Tree pose"*<br>Male P10 *"I'm the downward dog"*<br>Female P8 *"We went up like a seal"* |
| | | Pupil describes also doing yoga at home<br>7 pupils<br>3 girls<br>4 boys | Female P7 *"In the house… I do it myself"*<br>Male P10 *"I do yoga at my house"*<br>Male P11 "I do it both. School and my house" |
| | Views regarding what yoga entails, and who is more suited to, engaged with, or better at, yoga | Teacher comments on the non-competitive nature of yoga<br>5 teachers | Ppt 6 *"There's never been kind of a competitive streak, which is really when you say it like that you think, oh, actually because if they're out in the field doing their sports day, or their races, straight away it's there, but actually no, it's never really shown... They know that it's the team effort. It's for us, and I think that's probably why it kind of it doesn't have that competitive element maybe"* |
| | | Teacher observes that boys/ male pupils sometimes are averse or reluctant to join in with school-based yoga<br>5 teachers | Ppt 5 *"Like one of the children* [a male pupil], *wasn't really engaging particularly well with it…I don't think they felt comfortable doing it…I think it was just he didn't want to do it in front of other people and didn't like doing it. I think he saw it a bit like a dance, that sort of thing, and just didn't want to do it"* |
| | | Pupil says that yoga is 'for' their gender<br>10 pupils<br>5 girls<br>5 boys | Male P3 *"It's, it's boys"*<br>Male P1 *"Only for boys"*<br>Female P12 *"Girls"*<br>Female P10 *"Girls* |
| | | Pupils say that yoga is not for any particular gender or subset of the population<br>9 pupils<br>4 girls<br>5 boys | Male P5 *"Both – it's boys and girls"*<br>Male P11 *"Yeah, they're both the same. Both are better…Yeah"*<br>Female P2 *"Girls and boys"*<br>Female P7 *"Same"* |

This, perhaps, is evidence that these children hold some gender stereotypical beliefs about physical strength (for example, *"girls aren't usually as strong as boys"*, Male P9), as well as flexibility and dedication to practice (for example, *"Girls are, are more flexible than boys"*, Female P11; *"Girls take more practice"*, Female P8). However, nine pupils (five boys and four girls) made no reference to any gender being more able, or better suited to yoga:

*"Girls and boys"* (Female P2)

*"Same"* (Female P7)

*"Both – it's boys and girls"* (Male P5)

*"they're both the same…everyone"* (Male P7)

*"Yeah, they're both the same. Both are better…Yeah. Everyone in the world"* [both arms outstretched] (Male P11)

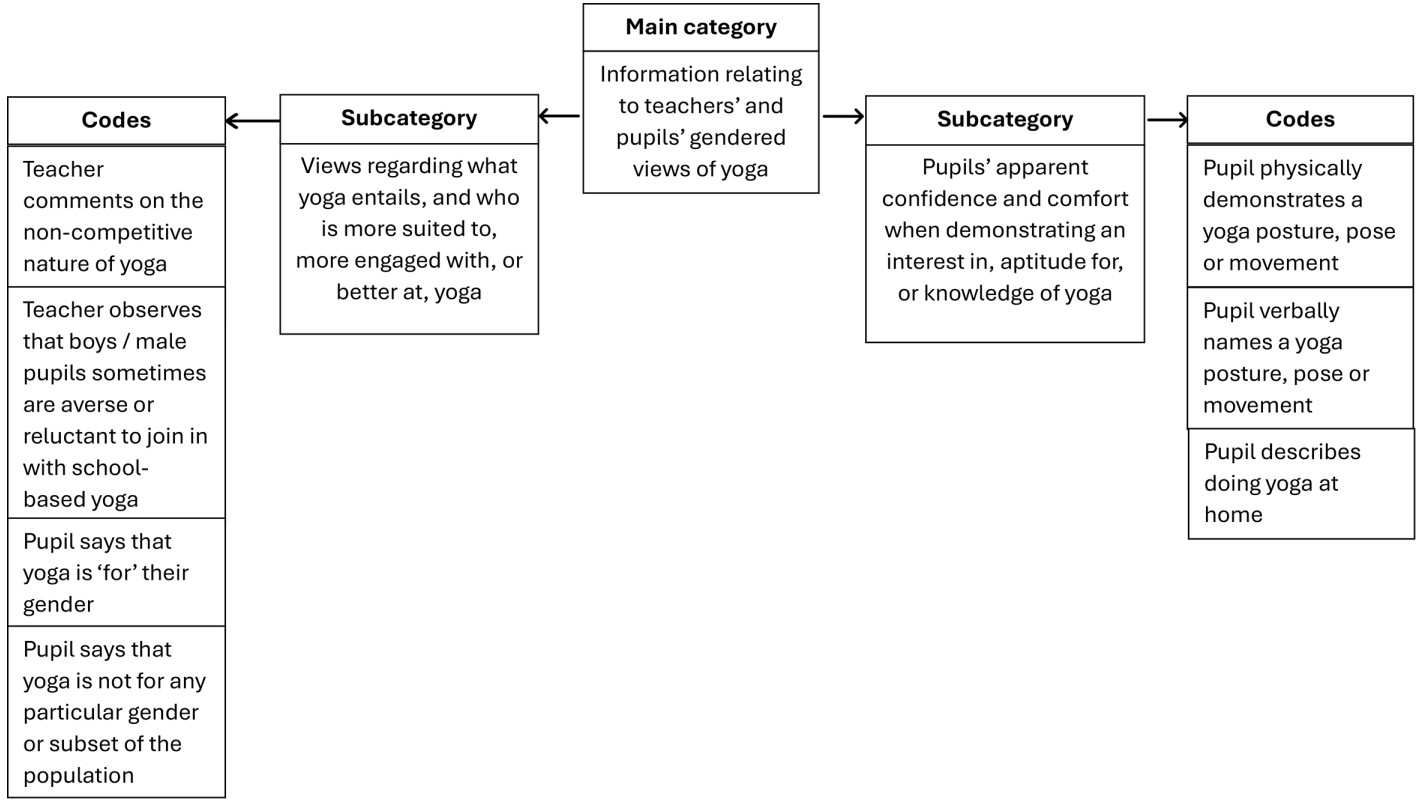

**Fig 1. Graphic summary of the QCA coding frame for selected pupil and teacher data.**

According to GST, children's knowledge of gender stereotypes begins prior to age two [75] and as this knowledge is accrued in the early years, children typically apply 'the rules' quite rigidly (for example, football is <u>only</u> for boys) [76]. However, from around six years of age, children's acceptance of gender stereotypes as correct or fixed begins to decline and more gender stereotype flexibility emerges [77], and this perhaps is reflected in these pupils' data – that any potential gendering of yoga is far from unanimous and is not considered accurate or appropriate by the majority of these five- and six-year-old pupils.

Alternatively, these pupils may have no consensus gendered view of yoga because they have had only minimal exposure to the 'yoga is a female/ feminine activity' stereotype. For instance, the most recent Sport England activity data tables do not include yoga in the 55 most popular physical activities as chosen by KS1 children in English schools [78], thus KS1 children generally are likely to have only limited experience with yoga as well as the gendering of this activity, and as GST explains, gender stereotypical schemas and stereotype congruent behaviour are fostered by increased exposure to gender stereotypes [79,80]. Indeed, while yoga marketing for adults and teenagers strongly targets women [81], commercial yoga resources for young children in the Key Stage 1 age-range (such as Cosmic Kids Yoga [82], Yoga Bugs [83], Salamander Yoga [84]) are more gender neutral, for instance making use of cartoons/ illustrations of characters and animals of indeterminate sex and gender. As such, it would be enlightening to gather more data on this topic from Key Stage 2 children (seven to 11 years of age) and Key Stage 3 children (11–14 years of age) to identify if/ when young children begin to formulate a more consistent gendered view of yoga.

### Teacher's views on yoga and gender

**Male aversion or reluctance to engage with school-based yoga.** When asked if any pupils appear less engaged or less motivated during yoga activities, five teachers spoke about some male pupils sometimes being less willing or enthusiastic to take part in yoga than their female peers. For example, "*I mean some of the boys weren't brilliant at first with joining in*" (Ppt 2); "*Little boys, like boys… just find it a little bit too much, you know*" (Ppt 4); "*The ones that I would say maybe don't find – they give it a go, but they're not as engaged as the other ones, erm, it's some of the boys*" (Ppt 6).

When discussing possible reasons for this, one teacher referenced flexibility – stereotypically a more feminine skill: "*I mean a few of them* [boys] *really struggled with things like touching their toes or, you know, doing the stretches*", (Ppt 2), and two teachers spoke about male pupils perhaps feeling self-conscious:

> "*Some boys just aren't happy to, it's just, you know, they struggle, they're not happy, they feel like they're very much on show*" (Pp 1)

> "*Like one of the children* [a male pupil]*, wasn't really engaging particularly well with it…I don't think they felt comfortable doing it…I think it was just he didn't want to do it in front of other people and didn't like doing it. I think he saw it a bit like a dance, that sort of thing, and just didn't want to do it*" (Ppt 5)

The fact that all but one teacher reported that male KS1 pupils sometimes appear resistant to engage in yoga is interesting, and to the best of our knowledge has not been reported elsewhere. However, this finding is perhaps unsurprising when we consider adults' gendered views of yoga being a female/ feminine activity [85], alongside evidence that primary schoolteachers' gender beliefs influence their expectations and perceptions of pupils' physical aptitude [48] and behaviour [46,86]. Indeed, GST would suggest that teachers' gender schemas influence the learning environment as teachers facilitate and reinforce children's gender-typed behaviours [87] and thus influence their pupils' motivation and skills [88].

Moreover, teachers are found to use gendered pedagogies for a variety of subjects and topics [89], including physical education [90] and social and emotional learning, where girls are coerced into demonstrating their learning, and boys are allowed to approach the lesson more passively [91]. Yoga, which aligns well with UK educational policies and frameworks for supporting pupil wellbeing and resilience [92], is sometimes used to support children's social-emotional development in school [93,94], complementing a school's PSHE (Personal, Social, Health and Economic) curriculum [95]. As female pupils are generally perceived as more engaged and responsive during school-based social-emotional learning activities than their male peers [96], teachers may have different expectations and implementation behaviours for their female and male pupils during such activities, which in turn may affect pupil engagement and responsiveness [97]. Future research to explore this issue in greater depth would be valuable.

There is also some discrepancy between the teachers' and pupils' accounts in this study, whereby teachers say their male KS1 pupils can be reluctant to engage in yoga, but male and female KS1 pupils appear equally confident and comfortable demonstrating their interest in, aptitude for, and knowledge of yoga, and seem to hold no consensus view of yoga being a feminine/ female activity. While research into this topic is scarce, there is one study which has evidenced a similar mismatch between teachers' perceptions of activity preference according to pupil gender, and male and female pupils' attitudes towards activities [98], and further examination of this issue would be valuable. Indeed, teachers' gender stereotypes are pervasive in primary education [40], influencing teachers' expectations of female and male pupils, and reinforcing perceived gender differences [47]. Such gender bias and stereotyping can impact pupils' self-concept [99], social and emotional skill development [44], and wellbeing [100]. Thus, evidence of disparity between teachers' and pupils' gendered views of school activities is important and warrants further attention.

Research also demonstrates that male pupils tend to derive more enjoyment, excitement [101] and pride [102] from physical activity in school than female pupils, who report greater discomfort [103], alienation [104], embarrassment and intimidation during participation [105]. However, such research has generally involved more traditional, competitive, team

sports, so to explore whether the non-competitive nature of yoga is of relevance, teachers were asked about any competition between their pupils during yoga activities. In response, only one teacher said they had observed this: "*Yes, there is always competition. But again, each child knows and feels, you know they're a champion at something, so that little bit of competition, I think, is valid*" (Ppt 1). The other five teachers all denied observing any competitive element during yoga with their KS1 pupils. For example: "*It* [competition] *is not something I've consciously noticed…no, I've not. That's not something I would say I have witnessed personally*" (Ppt 4).

One teacher felt that the lack of competition during yoga related to it being seen as a personal, individual activity:

"*I think it was very much, each in their own individual space. Because they weren't working together with a partner or as a team, I don't think they were ever really aware of that…that there were other people next to them that they could compare themselves with necessarily*" (Ppt 3)

Whereas another teacher spoke about yoga being a collective "*team effort*" which fosters cohesion rather than competition:

"*There's never been kind of a competitive streak, which is really when you say it like that you think, oh, actually because if they're out in the field doing their sports day, or their races, straight away it's there, but actually no, it's never really shown... They know that it's the team effort. It's for us, and I think that's probably why it kind of doesn't have that competitive element maybe*" (Ppt 6)

The absence of competition from yoga is perhaps relevant for male pupil participation. For instance, there is evidence of gender differences in competitiveness [106,107] such that males are more likely than females to seek out activities involving competition [108] and derive greater enjoyment from competition than females [109]. Also, although teachers aim to be fair and impartial – striving to provide equitable learning opportunities for all [48], teachers generally expect male pupils to be more competitive than their female peers [96] and are more likely to encourage competition in their male than their female pupils [42]. Thus, in recognising yoga as non-competitive, these teachers may presume their male pupils will be less engaged than their female peers.

Moreover, teachers tend to have lower expectations for boys' engagement and behaviour in school in general [110,111], so the teachers in this study may describe their male pupils as less engaged in yoga because they consider they are less engaged in classroom activities overall. However, little is known about how teachers' gender influences their expectations of male and female student engagement [112] and as the current study involves only teachers who identify as female, further research into this issue is required.

## Conclusion

Yoga tends to be considered a feminine/ female activity [15] but there has been minimal consideration of whether young children share this gendered view. This paper aimed to provide some understanding of male and female KS1 pupils' attitudes and opinions of school-based yoga, as well as KS1 teachers' perceptions of male and female pupil engagement with this activity. The data were considered according to GST which explains that a person's gender stereotypical schemas are created and promoted by exposure to gender stereotypes [79,80].

Collectively, the pupils' data indicate no shared consensus gendered view of yoga being a female/ feminine activity. Male and female pupils were equally confident and comfortable demonstrating their knowledge and physical aptitude for yoga and were equally likely to describe doing yoga at home as well as at school. In contrast, five KS1 teachers noted their male pupils sometimes appear averse or reluctant to join in with school-based yoga. To the best of our knowledge, such divergence between teachers' perceptions of activity preference according to pupil gender, and male and female pupils' attitudes towards activities is evidenced on only one other study to date [98].

 

Although strategies are available to help teachers promote parity and equality between genders in school-based physical activity [112], typically these relate more to standard physical education classes involving competitive, team sports, thus an expansion of such work to include activities such as yoga – which is becoming more prevalent in schools [29] – would be valuable.

## Limitations

The authors acknowledge that the current study significantly oversimplifies the concept of gender by depicting it as binary. This was not the authors' intention, but rather it was the best reflection of the language used by the study participants when describing and discussing gender. Recent work has explored gender fluidity and non-binary gender identities in young children [113,114] and we recognise and respect that young children's understanding of gender may well be more expansive and diverse than the simple binary notions presented in this paper.

Furthermore, all teacher participants identified as female, and although the population of teaching staff across England is predominantly female (76% identify as female according to the UK Government's most recent figures [115]), for a more comprehensive understanding of this topic, further engagement and research is needed with teachers whose gender identity is not female.

Also, there is no universally agreed definition of yoga [116], which is a term used to describe a wide range of diverse practices. As such, there are inherent difficulties when conducting research involving yoga [117], and as this study included no assessment of participants' definition and understanding of the term 'yoga', no claims can be made about the specific practices and activities the participants were describing.

Moreover, while the current study considered pupils' verbal contributions and physical demonstrations, it included no direct observation of KS1 pupils' engagement with school-based yoga, so there are no direct measures to support or refute the participants' accounts. Indeed, during the task, pupils may have responded according to what they thought was expected by the adult researcher [118] rather than feeling able to more honestly describe their opinions of whether/ how yoga is gendered, so future research involving triangulation, for instance combining observational with further qualitative data would refine our understanding of this issue. In addition, the study's findings and conclusions were formed without member checking or input from an external 'critical friend' (i.e., an individual outside the research team). Employing these measures in future qualitative research would improve objectivity.

Finally, as well as being perceived as a predominantly 'female'/ 'feminine' activity [119], yoga also is commonly associated with, and practiced by, those in or above the middle classes [120,121]. Thus, further research is needed with teachers and schoolchildren across the UK to explore this topic in areas of differing socio-economic statuses/ deprivation levels.

## Supporting information

**S1 File. QCA procedure and coding frame.** This document provides details of the Qualitative Content Analysis (QCA) procedure used in this study, as well as the full QCA coding frame for the teachers' and pupils' data.
(PDF)

## Acknowledgments

The authors gratefully thank the teachers and pupils who kindly participated in this study.

## Author contributions

**Conceptualization:** Katie Wilkin, Georgia Allen-Baker.

**Formal analysis:** Katie Wilkin, Georgia Allen-Baker.

**Investigation:** Katie Wilkin.

**Methodology:** Katie Wilkin, Georgia Allen-Baker.

**Supervision:** Claire Thornton, Georgia Allen-Baker.

**Validation:** Katie Wilkin.

**Visualization:** Katie Wilkin.

**Writing – original draft:** Katie Wilkin.

**Writing – review & editing:** Claire Thornton, Georgia Allen-Baker.

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
