## [Decision Letter · Decision Letter 0]

12 Jan 2026

Dear Dr. Wilkin,

Thank you for submitting your manuscript to PLOS ONE. After careful consideration, we feel that it has merit but does not fully meet PLOS ONE’s publication criteria as it currently stands. Therefore, we invite you to submit a revised version of the manuscript that addresses the points raised during the review process.

We look forward to receiving your revised manuscript.

Kind regards,

Simone A Tomaz, Ph.D.

Academic Editor

PLOS One

Journal Requirements:

2. Please include captions for your Supporting Information files at the end of your manuscript, and update any in-text citations to match accordingly. Please see our Supporting Information guidelines for more information: http://journals.plos.org/plosone/s/supporting-information .

Additional Editor Comments:

Dear author(s). Thank you for submitting this manuscript. It was a bit of a struggle to get a second reviewer and we somehow got lucky with 2 keen invitees in December, hence having 3 reviewers. I recognise that this increases the revision load but after reading through all 3 reviews, there is some overlap between reviewer comments and I am happy for you to highlight where there is overlap in your response to avoid duplication of work. I look forward to receiving your revised manuscript. Happy new year!

Reviewers' comments:

Reviewer's Responses to Questions

**Comments to the Author**

1. Is the manuscript technically sound, and do the data support the conclusions?

Reviewer #1: Yes

Reviewer #2: Partly

Reviewer #3: Partly

2. Has the statistical analysis been performed appropriately and rigorously?

Reviewer #1: N/A

Reviewer #2: N/A

Reviewer #3: N/A

3. Have the authors made all data underlying the findings in their manuscript fully available?

Reviewer #1: Yes

Reviewer #2: Yes

Reviewer #3: No

4. Is the manuscript presented in an intelligible fashion and written in standard English?

Reviewer #1: Yes

Reviewer #2: Yes

Reviewer #3: Yes

Reviewer #1: Introduction

Really well written introduction section. Some suggestions below:

Because the research is UK based, it could be helpful to link to some UK stats on yoga participation: Cartwright, T., Mason, H., Porter, A., & Pilkington, K. (2020). Yoga practice in the UK: A cross-sectional survey of motivation, health benefits and behaviours. BMJ open, 10(1), e031848.

Line 108-109 – when saying ‘much evidence’ can you link the reader to review level evidence rather than a single study? If no review level evidence is present, then is there more than one study you can refer to, otherwise use more cautious wording.

Methods

Really excellent to see and I commend the authors section on positionality. My only critique would be that it would have been helpful to have utilised a ‘critical friend’ from outside of the research team for objectivity – although I know that this can be difficult to access with limited researcher capability/capacity! Maybe something to consider for future qualitative research.

Tables 1-2 – is it necessary to have the locations of the schools/teachers? If not necessary, I would remove to protect anonymity. You could classify the schools with number IDs and add ‘primary/infant’ etc.

Line 188-189 – Some additional detail on the assent processes would be helpful with the children being so young, how did you know they understood what was being asked of them?

Line 191 – How would you define contribution and ‘engaged’ in this setting? Did you have any tactics to involve those who may not have felt as comfortable engaging in the more traditional sense?

Can you add some detail on justification of sample size for your study?

Line 266 – not sure this reference is in the right place?

Findings

A really thoughtful section and very interesting to read the contrasting opinions/perceptions between students and teachers.

Line 271-273 – Very hard to understand the meaning of this without knowing in-depth the method used – the figure helps slightly but I wonder if there is another way to describe this. And quite interesting that the analysis of the pupils and teachers has been combined – can you detail why. Was it to avoid duplication or did the data complement each other?

As you are discussing the findings with existing literature in the findings section, should you rename it ‘findings & discussion’?

Line 321-326 – Suggestion: because you are repeating a lot of the quotations to discuss, would there be a better way to format this? To avoid duplication could you present a quote and discuss, instead of presenting all quotes, then discussing them again?

Line 344 onwards – agree with this and would be interesting to discuss how yoga is marketed in this age group compared to other lifespan groups. For example, Cosmic kids is considered quite gender neutral and geared towards younger kids versus Studio You which is powered by This Girl Can.

Conclusion

This is written quite differently from typical journal articles and seems slightly repetitive with duplication of what we have just read including the references, rather than summarising the main takeaways. Is there a way to condense this section and make it slightly more punchy – the ‘so what?’ of the research needs to come out a bit more.

Limitations

Good section – only suggestion would be that you could perhaps explore the heterogeneity of yoga and the difficulties of research because of this.

Reviewer #2: 1. General assessment

This manuscript presents a timely and relevant qualitative study examining Key Stage 1 teachers’ and pupils’ gendered perceptions of yoga in the North-East of England. The topic contributes meaningfully to the fields of gender studies, educational psychology, and school-based wellbeing interventions.

The research is grounded in GST and utilises qualitative content analysis to investigate the gendering of yoga in early education. The study is methodologically sound in many respects, adheres to ethical standards, and presents original findings not previously published elsewhere.

However, the manuscript would benefit from several important revisions to strengthen clarity, transparency of analysis, and alignment between claims and evidence. Specific attention is needed in the areas of theoretical elaboration, analytic transparency, and framing of conclusions.

2. Title and framing

Title Revision Required: The current title foregrounds the teacher’s view ("He saw it a bit like a dance...") and does not adequately reflect the dual focus of the study on both teachers and pupils. It risks misrepresenting the scope, which includes children’s perspectives. A more neutral and inclusive title is advised, ideally reflecting the mismatch between teacher and pupil perspectives.

3. Clarity and Transparency of Methods

Missing content analysis table

The inclusion of a table summarising QCA categories and subcategories is essential. While a graphical coding frame (Figure 1) is referenced and supplementary materials are noted, the main manuscript would greatly benefit from a table displaying key categories,codes, subcodes, and example quotes, as per best practices in qualitative research reporting.

Interview protocol

Although semi-structured interviews are described, the article would be enhanced by including example interview questions, particularly those that elicited gender-related insights.

Analytic elaboration needed

The explanation of the QCA process (p. 11,12) could be expanded. More detail on how consensus was reached, how data saturation was considered, and how reflexivity was addressed during interpretation would strengthen methodological robustness.

4. Theoretical and contextual clarifications

Gender Schema Theory

The manuscript introduces GST appropriately, but its application could be more explicitly connected to the findings. For example, the claim that teachers' gender schemas shape pupil engagement (lines 387-389) is important and should be more deeply theorised, perhaps with references to how schema operate implicitly in classroom practices.

Personal experiences of researchers

The lead author's positionality (lines 124-131) mentions personal experiences with yoga/mindfulness but lacks elaboration. Were these experiences positive, challenging, or mixed? Greater context would clarify potential influences on the analytic lens.

Teacher gender: All participating teachers identified as female. The manuscript acknowledges this in Limitations but does not discuss its possible influence on pupils' participation or perception of yoga. This could be more directly addressed earlier in the discussion.

Ethnic and cultural context: Given the North-East England context and its socioeconomic profile (lines 144–149), a brief comment on ethnic or cultural diversity of the pupils (if available) would help assess how representative or generalisable the findings may be.

5. Interpretation of findings

Assumption of gendered intent in teaching yoga

The manuscript asserts that teachers may be using yoga to support social-emotional learning and that they may do so differently for boys and girls. This is a compelling claim, but it requires either clearer evidence from the data or greater support from existing literature.

Mismatch between teacher and pupil perspectives

The study finds that teachers perceive boys as less engaged, while pupils themselves (both genders) report enjoyment and participation. This is a critical insight that warrants further exploration, especially in light of potential adult gender schemas projecting onto pupil behaviour.

7. Additional Suggestions

WHO definition of gender

This appears late in the paper (line 166). It would be more appropriate to introduce the operational definition of gender earlier, ideally in the introduction or theoretical framework.

Limitations Section

Strong and honest. However, it could be further improved by acknowledging the absence of member checking or triangulation.

8. Conclusion and Recommendation

This manuscript offers valuable insights into a neglected area: gendered perceptions of yoga in early childhood education. However, several key elements need clarification or elaboration to meet the standards for qualitative research publications.

I recommend major revisions in accordance with the comments above.

Reviewer #3: Explanations about the 1st Comment to the Author:

The study explores an interesting gender-based perception of an emerging physical activity at schools: yoga. The conclusions presented are clearly derived from the collected data. However, I have some second thoughts regarding the conclusion that "any potential gendering of yoga is not fixed and is not considered accurate or appropriate by the majority of these five- and six-year-old pupils" (Line 341-342). It is not obvious that the age of the pupils who express a non-stereotypical gendered view of yoga is around 6 years, as the Gender Schema Theory suggests on Line 337-339 (Table 2, page 8 offers a participant-age spectrum from infant-school up to first-school age). Moreover, it seems that 9 out of 23 pupils say that yoga is not for any particular gender or subset of the population. On the contrary, 10 out of 23 pupils say that yoga is ‘for’ their gender. That leads me to question that any potential gendering of yoga is not fixed and is not considered accurate or appropriate by the majority of these five- and six-year-old pupils (Line 341-342). Complementary to this conclusion, the author states that only five pupils in this study (one male, four female) expressed gender-based stereotypical beliefs of girls’ and boys’ physical abilities in relation to yoga, while five male and five female pupils stated that yoga is exclusively ‘for’ their gender (Line 462-465). As the data analysis is based only on the verbally expressed gender-based stereotypical beliefs by the pupils, it seems possible that more (non) gender stereotypical beliefs exist that were not communicated verbally by the pupils and are not taken into consideration. On the same note, it is stated that most pupil participants described yoga as “for everyone”, “boys and girls” (Line 473-474). I feel that yoga is still presented in a gender stereotypical way, with short verbal statements of yoga being for all, and I am reluctant to conclude that the majority shares this belief.

Explanations about the 3rd Comment to the Author:

In the Data Availability section, it is stated that all data are fully available without restriction. However, the data underlying the results presented in the study are available from the lead author via email. I am wondering whether the video recordings and transcriptions can be found in an open source. Also, if the pupil qualitative task materials will be utilized somehow, as this study's findings derive only from the verbal statements of the pupils, if I understood correctly. Regarding the pupil qualitative task materials, I am wondering whether a case of a pupil arose that s/he chose a figure non-identical to their sex/gender, but their verbal statements came from their gendered views (Line 208-209).

Additional Comments:

The Gender Schema Theory is the primary theoretical framework employed, but the author also mentions alternative theories (Line 118-120). It would be interesting to include the names of alternative theories in the text, not just the given citations.

It would be great to have an interpretation about the female and non-male teacher participation (Line 162-163). Also, a comment to be included about the lack of primary school participants (Table 2, page 8). Lastly, adding an explanation of the Ofsted rating as a note (Table 2, page 8) would be helpful.

It is truly valuable that the teachers in this study work at schools with disadvantaged youth. I would like to read a relevant comment on gendered views of yoga and disadvantaged youth (Line 148-149).

It was a little unclear how the video recordings help with the contribution assurance (Line 203-205).

Under the Teacher’s views on yoga and gender section, two opposing perspectives appear: yoga offers an individual space, versus yoga is seen as a team effort. Could these perspectives be interpreted from the GST perspective?

**Do you want your identity to be public for this peer review?** For information about this choice, including consent withdrawal, please see our Privacy Policy

Reviewer #1: **Yes:** Dr Niamh Hart

Reviewer #2: **Yes:** Lea Loncar

Reviewer #3: No

---

## [Author Response · Author response to Decision Letter 1]

29 Jan 2026

Please see 'Response to reviewers' attachment provided

---

## [Editor Report · Decision Letter 1]

10 Feb 2026

Gendered perspectives of yoga in the Key Stage 1 classroom:  qualitative content analysis indicates contrasting views of teachers and pupils

PONE-D-25-54686R1

Dear Dr. Wilkin,

We’re pleased to inform you that your manuscript has been judged scientifically suitable for publication and will be formally accepted for publication once it meets all outstanding technical requirements.

Kind regards,

Simone A Tomaz, Ph.D.

Academic Editor

PLOS One

---

## [Editor Report · Acceptance letter]

PONE-D-25-54686R1

PLOS One

Dear Dr. Wilkin,

I'm pleased to inform you that your manuscript has been deemed suitable for publication in PLOS One. Congratulations! Your manuscript is now being handed over to our production team.

Kind regards,

on behalf of

Dr. Simone A Tomaz

Academic Editor

PLOS One